# Selective lipid recruitment by an archaeal DPANN symbiont from its host

Su Ding [1,4] ✉, Joshua N. Hamm [1,4] ✉, Nicole J. Bale [1], Jaap S. Sinninghe Damsté [1,2] & Anja Spang [1,3]

The symbiont *Ca*. Nanohaloarchaeum antarcticus is obligately dependent on its host *Halorubrum lacusprofundi* for lipids and other metabolites due to its lack of certain biosynthetic genes. However, it remains unclear which specific lipids or metabolites are acquired from its host, and how the host responds to infection. Here, we explored the lipidome dynamics of the *Ca*. Nha. antarcticus – *Hrr. lacusprofundi* symbiotic relationship during co-cultivation. By using a comprehensive untargeted lipidomic methodology, our study reveals that *Ca*. Nha. antarcticus selectively recruits 110 lipid species from its host, i.e., nearly two-thirds of the total number of host lipids. Lipid profiles of co-cultures displayed shifts in abundances of bacterioruberins and menaquinones and changes in degree of bilayer-forming glycerolipid unsaturation. This likely results in increased membrane fluidity and improved resistance to membrane disruptions, consistent with compensation for higher metabolic load and mechanical stress on host membranes when in contact with *Ca*. Nha. antarcticus cells. Notably, our findings differ from previous observations of other DPANN symbiont-host systems, where no differences in lipidome composition were reported. Altogether, our work emphasizes the strength of employing untargeted lipidomics approaches to provide details into the dynamics underlying a DPANN symbiont-host system.

Members of the DPANN archaea, originally named after the initials of its first five identified groups (Diapherotrites, Parvarchaeota, Aenigmarchaeota, Nanoarchaeota, and Nanohaloarchaeota), are characterized by small cell and genome sizes[1]. Since their discovery, additional lineages such as Woesearchaeota[2] and Pacearchaeota[2], Huberarchaeota[3], Micrarchaeota[4], Altiarchaeota[5], Undinarchaeota[6], and Mamarchaeota[7] have been identified and incorporated into the DPANN superphylum. These archaea are widely distributed in diverse environments, including hypersaline lakes[8,9], marine[6,10] and freshwater[1,2] bodies, sediments[11,12], acid mine drainage sites[4], and hot springs[13]. Apart from Altiarchaeota, most DPANN archaea exhibit limited catabolic capabilities necessary to sustain a free-living lifestyle[2,14] and the majority of them are predicted to

rely on symbiotic interactions with other organisms[14,15]. Indeed, the few cultivated DPANN have symbiotic lifestyles, with representatives from three lineages (Nanoarchaeota, Nanohaloarchaeota, and Micrarchaeota) available in co-culture with specific host archaea[9,13,16–24].

Due to incomplete biosynthetic pathways for nucleotides, amino acids, and lipids, most DPANN representatives are predicted to be dependent on metabolites from their hosts. Our current knowledge regarding the identity of those metabolites and the molecular basis for the exchange and/or uptake of these compounds is limited. Two DPANN representatives (i.e., *Nanoarchaeum equitans* and *Ca*. Micrarchaeum harzensis) are thought to acquire lipids from their hosts (*Ignicoccus hospitalis*[25] and *Ca*. Scheffleriplasma hospitalis[21], respectively). Lipid

[1]Department of Marine Microbiology and Biogeochemistry, NIOZ Royal Institute for Sea Research, Texel, The Netherlands. [2]Department of Earth Sciences, Faculty of Geosciences, Utrecht University, Utrecht, The Netherlands. [3]Department of Evolutionary & Population Biology, Institute for Biodiversity and Ecosystem Dynamics (IBED), University of Amsterdam, Amsterdam, The Netherlands. [4]These authors contributed equally: Su Ding, Joshua N. Hamm.
✉e-mail: su.ding@nioz.nl; joshua.hamm@nioz.nl

analyses of pure host and symbionts as well as symbiont-host co-cultures have revealed no significant qualitative difference in the lipid composition profiles between those cultures, indicating that the process of lipid uptake from the host is non-selective. However, it remains unclear whether non-selective lipid uptake from host organisms is a common feature among the various DPANN representatives.

In this study, we conducted a comprehensive analysis of the lipidome of the DPANN symbiont-host system consisting of *Ca*. Nanohaloarchaeum antarcticus – *Halorubrum lacusprofundi*[9,26]. *Ca*. Nha. antarcticus has so far not been obtained in a stable pure co-culture with *Hrr. lacusprofundi* (long term co-cultivation results in loss of *Ca*. Nha. antarcticus or death of the culture) and must be maintained in an enrichment culture (CLAC2B) containing multiple *Hrr. lacusprofundi* strains along with a *Natrinema* sp.[9]. Recent work has shown that the instability of pure co-cultures may be due to *Ca*. Nha. antarcticus being a parasite that invades the host cytoplasm leading to host cell lysis[26]. *Ca*. Nha. antarcticus lacks identifiable genes encoding proteins involved in lipid biosynthesis and metabolism and is thus hypothesized to rely on lipids of its host for survival[9]. Through the investigation of the lipidome of *Ca*. Nha. antarcticus and its host, we aimed to (1) determine whether the lipids of *Ca*. Nha. antarcticus closely resemble those of the host or exhibit differences; and (2) assess potential changes in the host's membrane lipid composition upon infection by *Ca*. Nha. antarcticus. Notably, our analyses reveal that *Ca*. Nha. antarcticus selectively takes up a specific set of lipids from its host. Moreover, in co-cultures, the lipidome composition undergoes changes that are likely compensating for an elevated metabolic load as well as enhanced mechanical stress on the host membrane.

## Results

### Molecular network of lipidome in the *Hrr. lacusprofundi-Ca. Nha. antarcticus* system

To determine the lipidome composition of *Ca*. Nha. antarcticus and *Hrr. lacusprofundi*, pure *Ca*. Nha. antarcticus cells were harvested from the nanohaloarchaeal enrichment culture[9] CLAC2B and inoculated into pure cultures of *Hrr. lacusprofundi* in mid-exponential phase at a ratio of 1:10 (*Ca*. Nha. antarcticus to *Hrr. lacusprofundi* cells). Biomass of co-cultures and pure *Hrr. lacusprofundi* was harvested at regular timepoints covering mid- to late- exponential phase of culture growth, whilst pure *Ca*. Nha. antarcticus cells were harvested pre- and post- co-culture growth experiments (Fig. 1a, details in the Method section). To account for potential variation in lipid profiles between *Hrr. lacusprofundi* strains and the possibility for acquisition of lipids from the *Natrinema* sp. present in the enrichment, biomass from 5 additional isolated *Hrr. lacusprofundi* strains and an isolate of the *Natrinema* sp. were used as quality controls (QCs), harvested at mid-exponential phase and subjected to lipidomics. Additionally, to assess whether differences in the *Ca*. Nha. antarcticus lipid profile were due to differences in lipid abundances within the enrichment culture, biomass from the enrichment culture was harvested for lipidomic analysis.

Growth of cultures was assessed by optical density, qPCR measurements, and 16S rRNA-targeted FISH microscopy (Fig. 1b, c, Supplementary Figs. 2–5). Optical density readings indicated active growth of both pure *Hrr. lacusprofundi* and co-cultures with the latter exhibiting a slightly slower rate of increase in density than pure culture. The qPCR data showed growth of *Ca*. Nha. antarcticus with an initial doubling of 16S rRNA copy number in the first 12 h followed by an approximately 100-fold increase between 24 and 48 h. Despite lower optical density readings, co-cultures displayed slightly higher 16S rRNA copy numbers for *Hrr. lacusprofundi*. *Hrr. lacusprofundi* 16S rRNA copy number remained stable in both conditions across the 48 h incubation consistent with previously reported genome copy number dynamics reported for other *Halobacteriales*[27]. FISH microscopy revealed multiple stages of interaction between *Ca*. Nha. antarcticus and *Hrr. lacusprofundi* in co-cultures (Supplementary Discussion,

Supplementary Figs. 2–4). *Hrr. lacusprofundi* cells displayed statistically significant shifts in size across pure and co-cultures with a similar trend in reduction in average cell size over time in both cultures (Supplementary Fig. 5, Supplementary Data 1). Its cell shape also varied significantly but trends were different between cultivation conditions, i.e., pure cultures displayed a gradual trend towards increased circularity, whilst co-cultures displayed a shift towards more elongated rod-shaped cells between 12 and 24 h before increasing in circularity at 48 h though to a lesser degree than in pure cultures.

We analyzed the lipidomes of the samples obtained in our experimental approach with ultra high-pressure liquid chromatography coupled with high-resolution tandem mass spectrometry (UHPLC-HRMS[2]) and handled the data obtained with a recently established pipeline that provides a comprehensive analysis of the microbial lipidome in both complex environmental samples and laboratory cultures[28,29]. In total, 2533 distinct ion components with associated MS[2] spectra were extracted and employed to build up a molecular network (Supplementary Fig. 6). Within this dataset, 1773 ion components (70%) occurred in structure-similarity groupings in the molecular network, while 760 ion components (30%) existed as singletons (i.e., lacking structurally related counterparts). The MS[2] spectra of these ion components did not retrieve any match to any related archaeal lipids in the Global Natural Product Social Molecular Networking (GNPS) spectral library. Lipidome annotation remains a challenge in lipidomic studies, as public spectral databases are inadequately populated. Nevertheless, by comparison with literature[30–37], as well as tentative identification, we were able to annotate 246 likely archaeal lipids (Supplementary Fig. 6).

These identified 246 lipids can be classified into two primary groups (Fig. 1d). The first group consists of bilayer-forming glycerolipids based on (extended) archaeol (AR, consisting of two ether-bound isoprenoid chains with either 20 or 25 carbon atoms). These occur either as AR core lipids (8 species, likely biosynthetic or degradation intermediates) or with a polar head group. The latter comprise phospholipids like phosphatidylglycerosulfates (PGS; 13 species), phosphatidylglycerols (PG; 25 species), phosphatidic acids (PA, 3 species), and dimeric phospholipids such as phosphatidylglycerophosphate methyl ester (PGP-Me, 22 species), biphosphatidylglycerols (PGPG, 11 species), and cardiolipins (CL, 41 species), as well as non-phospholipids, including other sulfur containing lipids than PGS (S, e.g., sulphated diglycosyl 3 species), monoglycosyl (1 G, 2 species), diglycosyl (2 G; 2 species) archaeol. The second group comprises non-bilayer forming lipids like menaquinones (MK, 39 species), squalene, and bacterioruberins (15 species).

Additionally, we found 60 unknown species associated with various major archaeal lipid classes. Due to the limited information on MS[2] fragmentation, we were unable to deduce their complete chemical structures. Nonetheless, critical fragments indicated their affiliation with archaeal lipids. For instance, within the 1 G/2 G/S subnetwork, two unknown lipids were linked to 1G-AR and 2G-AR (Source Data Table), featuring parent ions with *m/z* of 1324.385 and 1306.355. In their MS[2] spectra, both lipids exhibited diagnostic fragment ions at *m/z* 653.681 and 373.368, corresponding to AR, with an assigned elemental composition (AEC) of $C_{43}H_{89}O_3^+$ and lysoAR glycerol with AEC of $C_{23}H_{49}O_3^+$, respectively. Although their exact structures remained elusive, these lipids were included in the overall statistical analysis.

### Structural diversity and specificity of lipidome in the *Hrr. lacusprofundi-Ca. Nha. antarcticus* system

To assess the similarity in lipidome composition between the host *Hrr. lacusprofundi*, the symbiont *Ca*. Nha. antarcticus, their co-cultures, and corresponding QCs across multiple time series and replicates, we performed a Principal Component Analysis (PCA) on the abundances of lipid species (Fig. 1e). The first two principal components (PC1 and PC2) accounted for 45.3% of the total lipid variance. The majority of the pure host *Hrr. lacusprofundi* cultures harvested at different time points

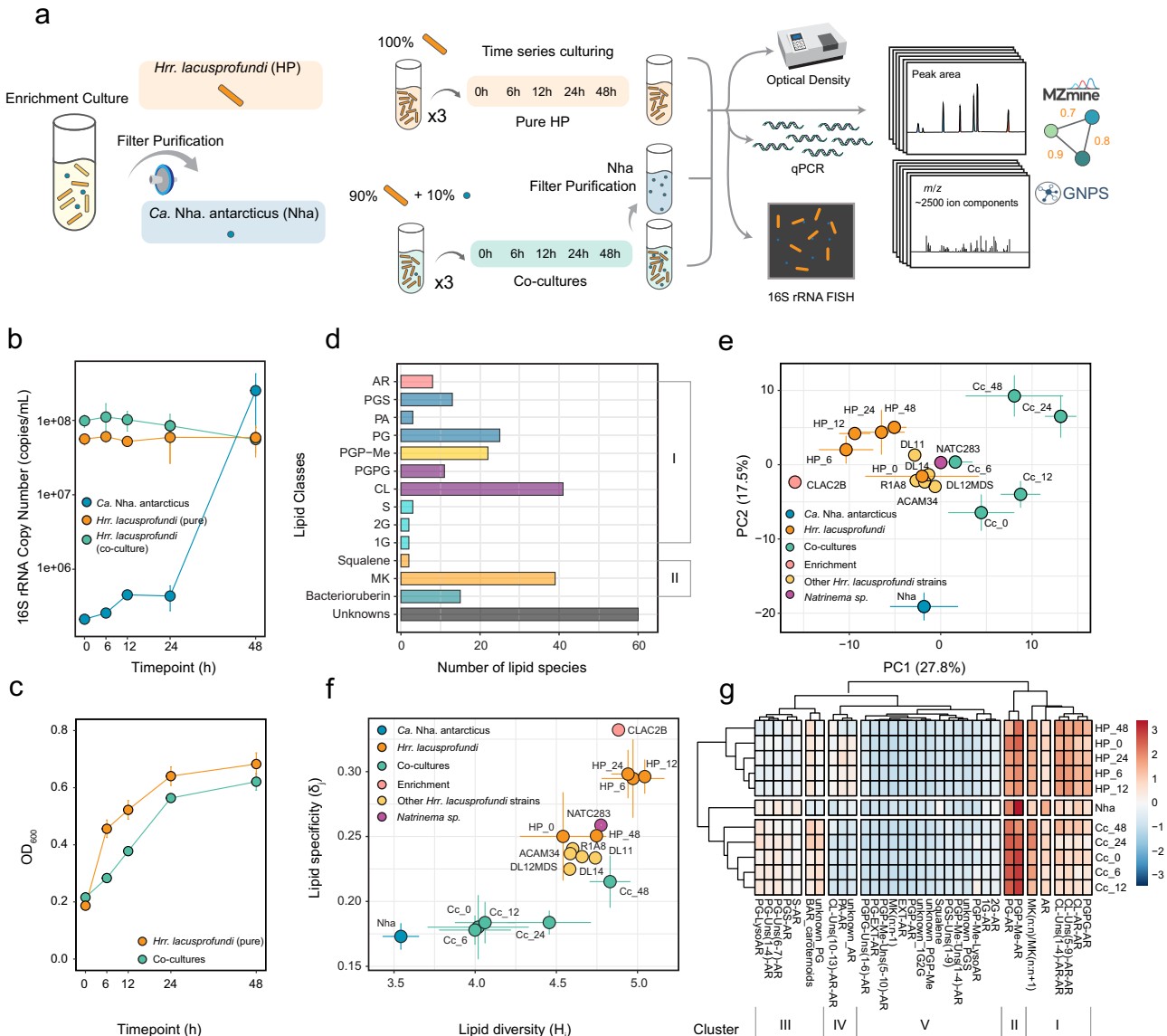

**Fig. 1 | Overview of the experimental design and the general lipidome composition in the *Hrr. lacusprofundi*-*Ca*. Nha. antarcticus system. a** Schematic overview of the experimental design. **b** qPCR-based growth measurements. Error bars show the standard deviation of calculated 16S rRNA gene copy number. **c** Optical density at 600 nm (OD$_{600}$) growth measurements. Error bars show the standard deviation of measured OD$_{600}$ values. **d** The number of individual lipid species in major lipid classes among all the samples. **e** Principal Component Analysis (PCA) based on the abundance of intact polar lipid species, showcasing the variance in general lipidomic features among distinct cultures or over varying culture durations. **f** Information theory analysis showing lipidome diversity ($H_j$ index) and specialization ($\delta_j$ index) based on the Shannon entropy of the lipidomic frequency distribution. Error bars in the data represent variability across replicates. **g** Hierarchical clustering heatmap depicting the distribution of major lipid classes among distinct cultures or over varying culture durations. The color bar on the right side represents Z-score normalization scale (ranges from −3 to +3 standard deviation (SD)). Sample abbreviations: *Ca*. Nha. antarcticus (Nha), *Hrr.*

*lacusprofundi* (HP), co-cultures (Cc), *Natrinema* sp. (NATC283), Enrichment (CLAC2B), other *Hrr. Lacusprofundi* strains (DL11, DL14, DL12MDS, R1A8, ACAM34). The sample abbreviation with a number stands for its culturing time, for example, CC_6 means co-cultures sampled at 6 h. In Fig. 1b, c, f, and e, data of Cc and HP cultures correspond to the mean values ± SD of three technical replicates. Nha has two replicates. Source data are provided as a Source Data file. Lipid abbreviations: archaeol core lipids (AR), phosphatidylglycerol (PG), phosphatidylglycerosulfate (PGS), phosphatidic acid (PA), phosphatidylglycerophosphate methyl ester (PGP-Me), biphosphatidylglycerol (PGPG), cardiolipin (CL)monoglycosyl (1 G), diglycosyl (2 G), archaeol lipids containing a sulfur-containing head group except for PGS (S), menaquinone (MK), an "extended" "archaeol chain", i.e., with a C$_{25}$ isoprenoid carbon chain (EXT-AR), unsaturation in the archaeol chain (uns). The two "n" in MK (n:n) stand for numbers of the isoprenoid unit in the side chain and unsaturation in the isoprenoid chain, respectively. MK(n:n-1) signifies one less double bond in the nth isoprenoid chain.

were closely clustered together, adjacent to the enrichment culture, indicating their high similarity. In contrast, the co-cultures formed a distinct cluster by scoring positively on PC1. The pure *Natrinema* sp. which is the third species isolated from the enrichment and five additional *Hrr. lacusprofundi* strains QC exhibited proximity to the host (depicted in Fig. 1a and Methods section), underscoring the robustness of the culturing experiment and lipidome analysis results.

Importantly, a clear separation of the lipidome of *Ca*. Nha. antarcticus, scoring negatively of PC2, from all other samples was observed.

To access the lipidome plasticity of the *Hrr. lacusprofundi*-*Ca*. Nha. antarcticus system, we also employed an information theory framework[38,39], quantifying lipidome diversity ($H_j$ index) and specialization ($\delta_j$ index) based on the Shannon entropy of the lipidomic frequency distribution. Upon plotting the lipidome specialization and

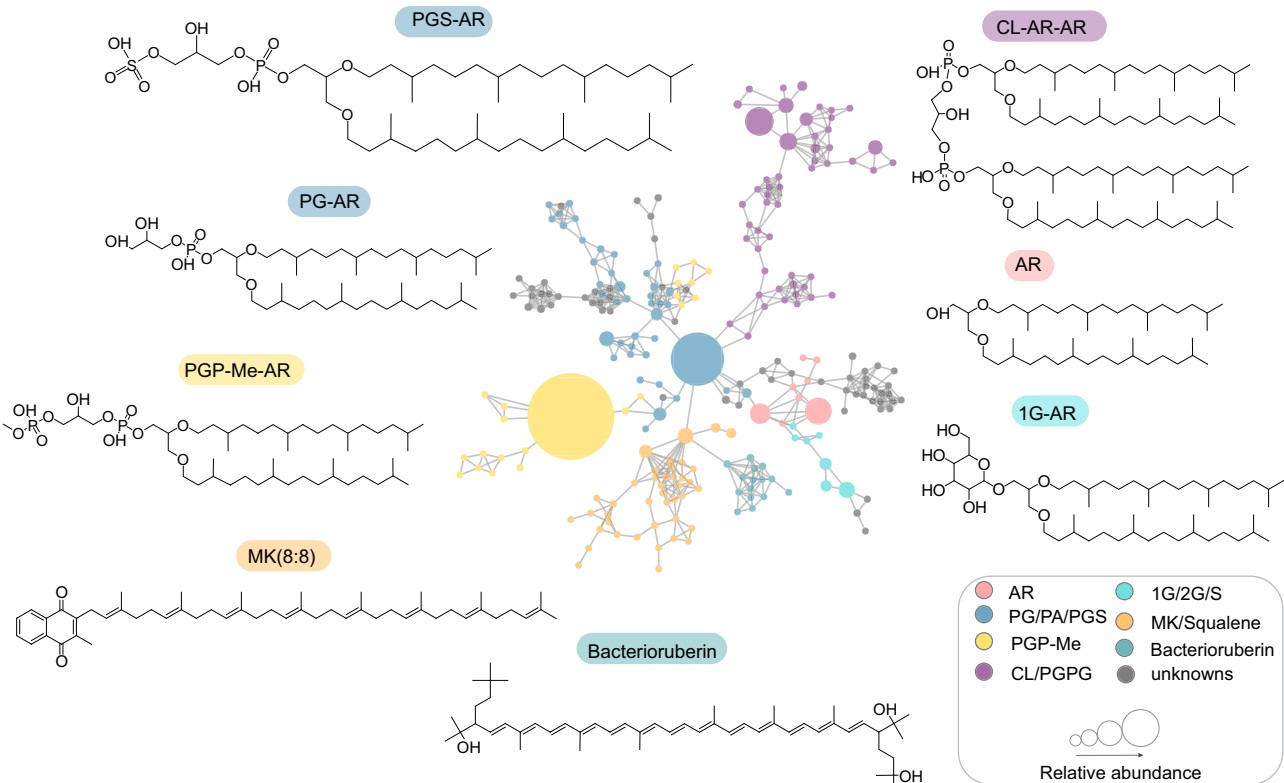

*Ca.* Nha. antarcticus (Nha)

PGS-AR · CL-AR-AR · PG-AR · AR · PGP-Me-AR · 1G-AR · MK(8:8) · Bacterioruberin

AR · 1G/2G/S
PG/PA/PGS · MK/Squalene
PGP-Me · Bacterioruberin
CL/PGPG · unknowns

Relative abundance

**Fig. 2 | Molecular network of the relative abundances of individual lipids of *Ca.* Nha. Antarcticus.** *Ca.* Nha. antarcticus was isolated from the co-culture after 48 h of growth. The nodes in the molecular network represent MS/MS spectra of ion components (lipids) which are connected based on spectral similarity. The sizes of the lipid nodes indicate their average relative abundance. The abbreviations of lipids are given in the caption of Fig. 1. The network presented here was a composite derived from merging the lipid subnetworks shown in Supplementary Fig. S6. Figure 2 offers an overview of the major lipid compositions of *Ca.* Nha. antarcticus, highlighting the significant difference in lipid types between our symbiont system and others that primarily consist of monolayer membrane tetraether lipids[21,25].

diversity of the *Hrr. lacusprofundi-Ca.* Nha. antarcticus system and QCs, we observed that the pure host *Hrr. lacusprofundi* and QC samples exhibited relatively high lipid diversity and specialization compared to the symbiont (Fig. 1f). In contrast, the co-cultures showed relatively low lipid diversity during the first 12 h of cultivation, which subsequently increased from 24 to 48 h, resulting in a more diverse lipidome profile which was comparable to the profiles of the pure *Hrr. Lacusprofundi* harvested at different time points. The enrichment cultures displayed the highest lipid specialization, while the symbiont exhibited the lowest lipid diversity and specialization, which may be the result of the symbiont's lack of identifiable unique genes responsible for lipid biosynthesis[9].

We further employed a hierarchical clustering heatmap to examine the distribution of approximately thirty major lipid classes in the host *Hrr. lacusprofundi*, the symbiont *Ca.* Nha. antarcticus, and their co-cultures (Fig. 1g). These lipid classes were categorized based on their polar head groups, degrees of unsaturation, and chain lengths. This method allowed us to evaluate the resemblance in lipid class profiles across the cultures and provided an overall comparison of culture similarity grounded in these lipid classifications. The thirty-three major lipid classes were grouped into five distinct clusters. The first cluster consisted of AR core lipids, MK with a high degree of unsaturation [e.g., MK(8:8) and MK(8:9), the number of the isoprenoid unit in the side chain: the number of double bonds], PGPG and CL with varying degrees of unsaturation. The second cluster comprised phospholipids PG and PGP-Me. The third cluster included PG unknowns, PG-LysoAR (which featured only one $C_{20}$ or $C_{25}$ chain) and

bacterioruberins. The remaining clusters comprised highly unsaturated phospholipids (e.g., CL with 10-13 double bonds), 1 G, 2 G, sulfur-containing lipids, along with several less abundant lipids and unknowns. The lipid class composition of the host sampled at different harvest times were similar to each other, as were lipid profiles of the co-cultures obtained at different harvest times. Consistent with the PCA analysis and characterization of the lipidome diversity and specialization, which are both based on the complete lipid species profile, the hierarchical clustering based on the major lipid class compositions also revealed that the host, co-cultures, and the symbiont differed from each other.

**Changes in the molecular network of the lipidome over time.** We next analyzed the variation in the lipidome compositions at the species level by integrating molecular subnetworks of various lipid classes (Fig. 2 and Supplementary Fig. 7) from the overall molecular networks (Supplementary Fig. 6), based on their relative abundances. For some lipid species, abundances displayed significant variation between both cultivation conditions and time points of the same cultivation condition (Fig. 3a). The relative abundance of PGP-Me-AR comprised 44% of total lipids in the symbiont, significantly higher ($P < 0.05$, Tukey's Honest Significance Difference test, Fig. 3a) than in the pure culture of the host (13-21%) and the co-culture (12-31%). In addition, bacterioruberins were underrepresented in symbiont biomass (0.2%) compared to both pure culture of the host (1.2-2.9%) and co-culture (1.1-5.2%) total lipids, irrespective of time. The pure *Hrr. lacusprofundi* culture also displayed a significantly different lipid profiles compared to that

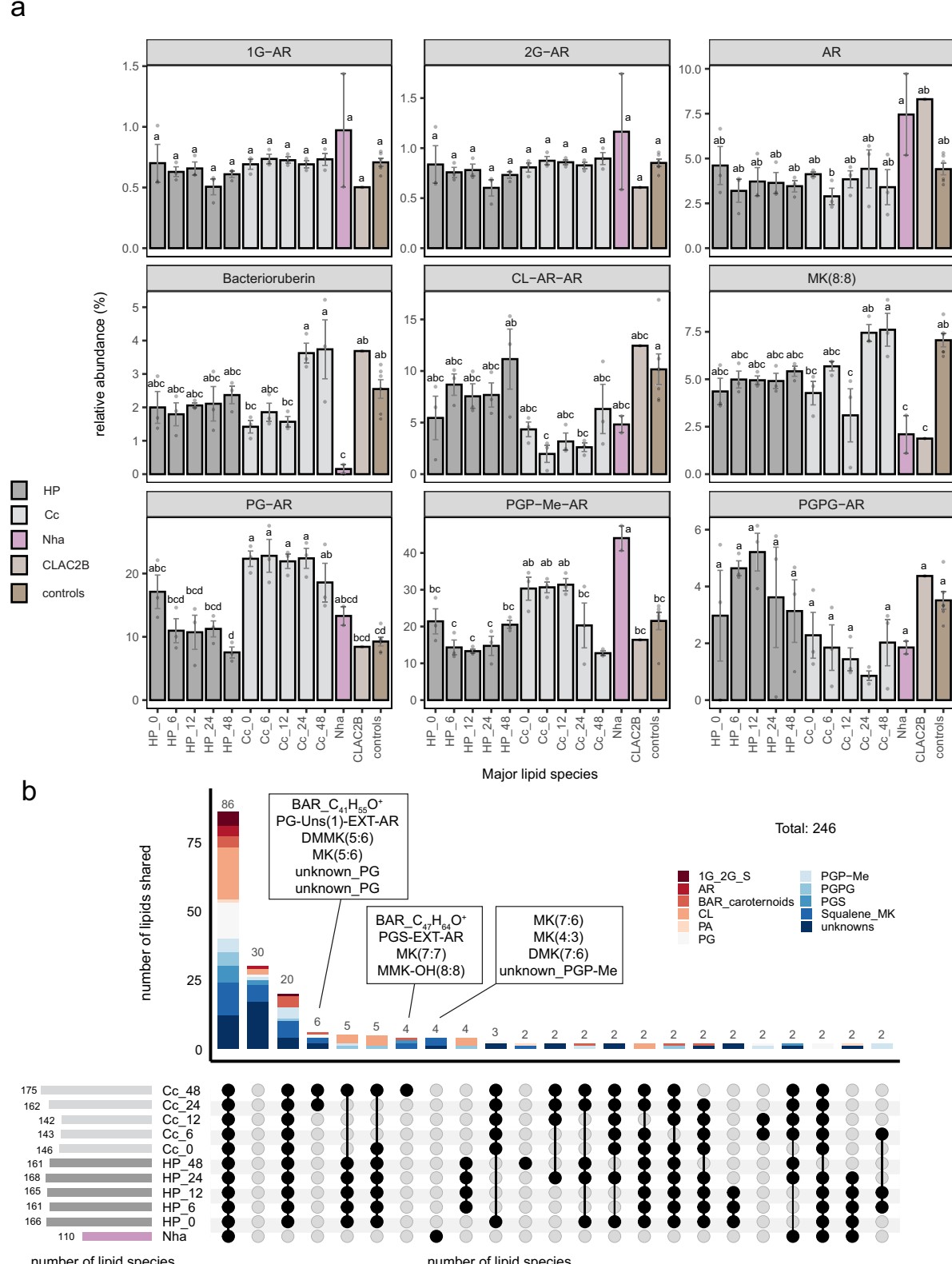

of the co-culture with statistically significant differences in the relative abundance of bacterioruberins, CL, PGP-Me, PG, and MKs of which only the relative abundance of PG was time-dependent (18.5–22.3% co-cultures, 7.5–17.1% pure *Hrr. lacusprofundi*). The remaining lipid classes displayed time-dependent differences in relative abundance, whereby CL was underrepresented in early-mid co-culture growth (6–24 h, 0.6–1.2% co-culture, 1.5–1.7% pure *Hrr. Lacusprofundi* culture), PGP-Me

was overrepresented in early – mid culture growth (0–24 h, 20.3–31.4% co-culture abundance, 13–21% pure *Hrr. lacusprofundi* abundance). In contrast, MKs (12.6–12.7% co-culture, 7.7–9.2% pure *Hrr. lacusprofundi*) and bacterioruberins (3.6–3.7% co-culture, 2.1–2.4% pure *Hrr. lacusprofundi*) both became significantly enriched in the late phases (24–48 h) of the co-culture biomass compared to the pure culture of *Hrr. lacusprofundi*. There were also statistically significant shifts in the

**Fig. 3 | The presence, absence, and changes in lipid composition in the *Hrr. lacusprofundi*-*Ca*. Nha. antarcticus system. a** The relative abundance of representative lipid species within the most dominant lipid classes. Statistical differences in lipid species among the samples were assessed using the one-sided Tukey's Honest Significance Difference test (TukeyHSD) with multiple comparisons, and results were visualized with the Compact Letter Display (CLD) ($P < 0.05$). **b** The shared lipid species across samples is illustrated through an UpSet plot[40,41]. A threshold of 0.01% relative abundance of total lipids was applied to determine the presence of a lipid in a specific sample; lipids with less than 0.01% of total lipid abundance were considered absent in that sample. The dark connected dots denote lipid species shared among these samples. Sample abbreviations: *Ca*. Nha. antarcticus (Nha), *Hrr. lacusprofundi* (HP), co-cultures (Cc) and Enrichment (CLAC2B). Data of Cc and HP cultures correspond to the mean values ± SD of three technical replicates. Nha has two replicates. *Natrinema* sp. (NATC283) and five other *Hrr. lacusprofundi* strains (DL11, DL14, DL12MDS, R1A8, ACAM34) are used as controls ($n = 6$). Lipid abbreviations: demethylmenaquinone (DMK)[85], methylmenaquinone (MMK), dimethylmenaquinone (DMMK). The representative lipid species are 1G-AR (*m/z* 832.760, $C_{49}H_{102}O_8N^+$), 2G-AR (*m/z* 994.813, $C_{55}H_{112}O_{13}N^+$), AR (*m/z* 653.681, $C_{43}H_{89}O_3^+$), Bacterioruberin (*m/z* 741.581, $C_{50}H_{77}O_4^+$), CL-AR-AR (*m/z* 1522.313, $C_{89}H_{183}O_{13}P_2^+$), MK(8:8) (*m/z* 717.560, $C_{51}H_{73}O_2^+$), PG-AR (*m/z* 807.684, $C_{46}H_{96}O_8P^+$), PGP-Me-AR (*m/z* 901.666, $C_{47}H_{99}O_{11}P_2^+$), PG-PG-AR (*m/z* 961.687, $C_{49}H_{103}O_{13}P_2^+$). PG, PG-AR, and unknown PG represent different lipid expressions. PG-AR signifies a PG headgroup linked to an archaeol core lipid, characterized by a lipid species with an *m/z* 807.684 and an elemental composition of $C_{46}H_{96}O_8P^+$. The term PG encompasses all PG lipid species, including PG-AR and PG-EXT-AR (which has an additional isoprenoid unit), and others. The term 'unknown PG' is used when the core lipid is unidentified. Source data are provided as a Source Data file.

degree of lipid saturation with both purified *Ca*. Nha. antarcticus cells and co-cultures displaying higher rates of saturation in bilayer forming glycerolipids when compared to pure *Hrr. lacusprofundi* cultures (Supplementary Fig. 8). The degree of saturation of MK also varied across samples with *Ca*. Nha. antarcticus and co-cultures in the late growth phase showing increased abundance of MK with one less saturation than the number of the isoprenoid unit in the side chain [e.g., MK(8:7), Supplementary Fig. 8].

**Lipid intersection in the *Hrr. lacusprofundi*-*Ca*. Nha. antarcticus system.** We next generated an UpSet plot[40,41] (Fig. 3b) to highlight the number of lipid species shared between samples, as well as those unique to either. Furthermore, this analysis provides insight into the number of lipid species potentially acquired by the symbiont from the host. As shown in Fig. 3b, the symbiont *Ca*. Nha. antarcticus contained 110 individual lipid species, while the pure culture of *Hrr. lacusprofundi* consistently contained approximately 165 lipid species. The co-cultures had around 140 lipid species in the first 12 h, progressively increasing to approximately 175 species by the end of the experiment at 48 h. This trend aligns with the observed lipid diversity $H_j$ index in Fig. 1f, suggesting that *Ca*. Nha. antarcticus incorporated only a limited number of lipid species from *Hrr. lacusprofundi*. Moreover, 86 lipid species across all major lipid classes were observed to be commonly present in all cultures. There were 20 lipid species exclusively found in both co-cultures and pure *Hrr. lacusprofundi* but absent in *Ca*. Nha. antarcticus. Notably, ten specific lipid species, were detected exclusively in the co-cultures between 24 and 48 h. These include two bacterioruberins, menaquinones with equal or additional unsaturation relative to their isoprenoid unit chain length [MK(n:n), MK(n:n + 1)], and specific phospholipids. Conversely, three menaquinones with one unsaturation less than their number of isoprenoid units [MK(n:n-1)] were uniquely present in *Ca*. Nha. antarcticus and the enrichment (Fig. 3b and Supplementary Fig. S9), specifically MK(7:6), MK(4:3), and demethylmenaquinone DMK(7:6). This underscores a notable distinction in lipid species composition among the host, co-cultures, and *Ca*. Nha. antarcticus.

## Discussion

Given the lack of key lipid biosynthesis pathways in the *Ca*. Nha. antarcticus genome (as shown in Fig. 4), the difference in lipidome composition observed between *Hrr. lacusprofundi* and *Ca*. Nha. antarcticus indicates that *Ca*. Nha. antarcticus selectively acquires specific lipid species from its host. These observations match with the lipid uptake behavior noted for archaeal viruses[42–45]. For instance, the *Sulfolobus* filamentous virus 1 selectively acquired lipids from its host *Sulfolobus shibatae* for survial[44]. However, our results contrast those of previous studies on two DPANN symbiont-host systems, which did not identify differences between symbiont and host lipid profiles[21,25]. These discrepancies may be attributed to natural differences between these distantly related symbiotic partnerships or could stem from

differences in the methodologies. Specifically, our untargeted lipidome approach provides higher resolution, enhancing the capacity to discern and compare lipid profiles, surpassing traditional methods. It will be important to address in future studies whether the application of this technique to other DPANN symbiont host systems may reveal a similar specificity in lipid uptake by those DPANN species. In terms of natural differences, unlike the other reported host archaea which primarily consist of monolayer membrane tetraether lipids, mostly glycerol dialkyl glycerol tetraethers[46], the cellular membranes of *Hrr. lacusprofundi* are exclusively formed by bilayer AR lipids[30,31,37]. The composition of bilayer-forming intact polar lipids in *Halobacteriales* stands as one of the most extreme instances of negatively charged membranes across the tree of life and is considered to be an adaptation to the high cationic environment[31]. For instance, *Halobacteriales* are the only archaea known for their unique capability to produce both PGP-Me and CLs[33,35,47–49]. This preference for bilayer membranes within *Halobacteriales* confers ecological advantages and the energy-efficient bilayer membrane structure observed in the host, *Hrr. lacusprofundi*, could potentially offer greater flexibility for *Ca*. Nha. antarcticus to selectively uptake specific lipids.

It has previously been shown that the distance between negative charges on the head group structure of CL reduces the efficiency of association with divalent cations such as $Mg^{2+}$ and favors association with monovalent cations such as $K^+$, whilst PGP-Me more efficiently associates with divalent cations[31]. The medium used for these cultivation experiments contains much higher concentrations of $Mg^{2+}$ compared to $K^+$ (514 mM combined $MgSO_4$ and $MgCl_2$, 39 mM KCl). In addition, experiments in reconstituted phosphatidylcholine-based membranes have shown that increased CL abundance reduces both membrane stability and the force necessary for membrane piercing[50]. Given that *Ca*. Nha. antarcticus lacks the capacity to regulate the composition of its membrane; it is plausible that an increased abundance of PGP-Me and a reduced abundance of CL, compared to its host (Fig. 3a), provides greater stability to the nanohaloarchaeal membrane under such high divalent cation concentrations, reducing the energy expenditure necessary for membrane maintenance.

The significant decrease in bacterioruberin abundance within *Ca*. Nha. antarcticus biomass compared to *Hrr. lacusprofundi* suggests a counter-selection against recruitment of bacterioruberins into the symbiont membrane (Fig. 3a). Bacterioruberin functions as an antioxidant, increases membrane rigidity, and is associated with rhodopsins within membranes of members of the *Halobacteriales*[51]. The exclusion of bacterioruberins from the nanohaloarchaeal membrane is in agreement with the prediction of lower oxidative stress in the symbiont due to its fermentative lifestyle or pressure to maintain higher membrane fluidity. In addition, whilst *Ca*. Nha. antarcticus possesses a rhodopsin gene, this is predicted to encode a sensory rhodopsin, in contrast to the rhodopsins that are present in the *Hrr. lacusprofundi* genome that likely generate proton gradients for ATP production and may necessitate more bacterioruberins. Consistent

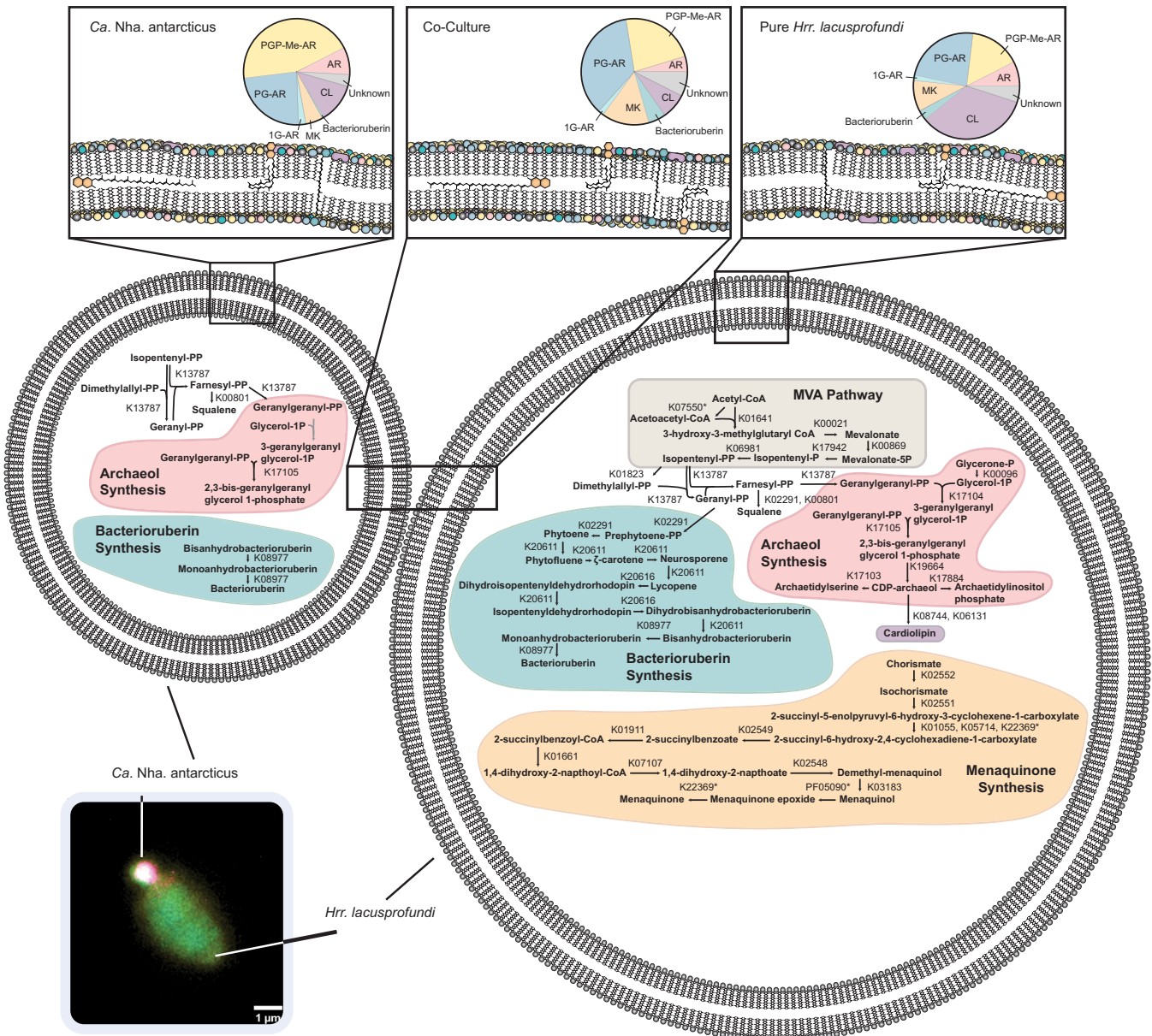

**Fig. 4 | A schematic figure showing lipid composition and biosynthetic pathways in *Hrr. lacusprofundi*, *Ca.* Nha. antarcticus, and co-cultures.** Lipid biosynthetic pathways were manually reconstructed using genome annotations of *Hrr. lacusprofundi* and *Ca.* Nha. antarcticus inferred using KEGG orthology (List of genes present in Supplementary Data 3 and 5). In some cases (marked by asterisks) the correct KEGG annotation could not be identified but a related enzyme that may carry out the reaction was present and is shown instead. Insets above cells show abundance of different lipid classes in each condition at 24 h. Localization of MKs within the bilayer is shown in two possible states in line with current uncertainty regarding the exact position MKs occupies within the membrane[31]. A representative 16S rRNA target FISH microscopy image as an example of the interactions between host and symbiont is included (Colors correspond to DNA: Blue, *Hrr. lacusprofundi*: Yellow, *Ca.* Nha. antarcticus: Magenta).

with oxidative stress playing a role in determining membrane composition, *Ca.* Nha. antarcticus also displayed a preference for MKs with increased degree of unsaturation, which has been suggested to operate more efficiently in hypoxic conditions[52]. MKs function as important carrier molecules within the electron transport chain during respiration, and have also been proposed to play a role in the regulation of membrane permeability through increased packing within the bilayer and oxidative stress through scavenging of free radicals[31]. Interestingly, for *Mycobacterium tuberculosis* increased unsaturation of MKs was proposed as an adaptation to intracellular environments, which are often lower in oxygen content[52]. Recently it was reported that *Ca.* Nha. antarcticus appears to invade host cells during the process of interaction[26] and the preference for increased MK desaturation may

similarly assist in survival during this stage of the symbiont's lifecycle. The contrasting metabolic strategies employed by this nanohaloarchaeum compared to its host may favor the selective acquisition of lipids seen in our data to maximize metabolic efficiency and survival.

In addition to the differences between *Ca.* Nha antarcticus and host lipid profiles, there were significant shifts in lipidome composition of the co-culture biomass and pure *Hrr. lacusprodundi*. This indicates a different membrane composition for host cells during co-culture. Initially co-cultures showed low lipid species numbers and lipid diversity during the first 12 h, which may partly be due to the limited lipid diversity of the symbiont *Ca.* Nha. antarcticus but also reflects changing lipid composition in the host *Hrr. lacusprofundi*. The subsequent rise in the number of lipid species and diversity in co-

**Table 1 | Detail of FISH Probes and PCR Primers used in this study**

| Name | Purpose | Target Organism | Sequence |
|---|---|---|---|
| Lacus-FISH-Cy3 | FISH Probe | *Hrr. lacusprofundi* | /5Cy3/TTATTACAGTCGACGCTGGTGAGATGTCCG |
| Nha-FISH-Cy5 | FISH Probe | *Ca*. Nha. antarcticus | /5Cy5/GTGTATCCCAGAGCATTCG |
| H_lac_qpcrF | qPCR Primer | *Hrr. lacusprofundi* | GGATTGTGCCAAAAGCTCCG |
| H_lac_qpcrR | qPCR Primer | *Hrr. lacusprofundi* | ACTCTCATGACCCGTACCGA |
| Nanohalo_qpcrF | qPCR Primer | *Ca*. Nha. antarcticus | ACTTAAAGGAATTGACGGGGG |
| Nanohalo_qpcrR | qPCR Primer | *Ca*. Nha. antarcticus | CATGCAGCTCCTCTCAGCG |

cultures over the following 36 h, along with the distinct lipid composition compared to the pure *Hrr. lacusprofundi*, suggests that *Hrr. lacusprofundi* modifies its membrane composition in response to interactions with *Ca*. Nha. antarcticus. Interestingly, lipid species enriched in *Ca*. Nha. antarcticus (e.g., PGP-Me) showed a decrease in abundance over the course of the time series in the co-cultures. In contrast, the diversity and abundance of other lipids, such as bacterioruberins and MKs, which were less prevalent in the nanohaloarchaeal membrane, showed an increase in abundance in the co-cultures. The increased abundance of bacterioruberins and MKs likely indicates a response from *Hrr. lacusprofundi* to an increase in metabolic load due to the presence of the symbiont. Similarly, bilayer-forming glycerolipids showed an increase in the rate of desaturation resulting in increased membrane fluidity and likely enhanced permeability for electron transport and therefore more efficient respiration[31,53].

Similar to *Ca*. Nha antarcticus, the CL abundance in the co-culture biomass also decreased, which likely reflects decreased abundance in both species' bilayers. As mentioned above, incorporation of CL within phosphatidyl-choline membranes reduces stability of the membrane and the energy necessary for membrane puncture[50] Reduced abundance within co-culture biomass may reflect an increase in membrane stability and resistance to mechanical stress. DPANN archaea that take up nutrients directly from host cells require access to their host's cytoplasm for nutrient acquisition and have been observed to form channels in host membranes as part of interactions[54]. Therefore, it seems possible that the decreased CL content in the *Hrr. lacusprofundi* membrane fortifies the membrane, which may impede predation by its symbiont (through increased resistance to membrane puncture) and/ or enhance survival of infected cells by reducing the chance of membrane destabilization.

Due to the nature of the *Hrr. lacusprofundi* – *Ca*. Nha. antarcticus system it was not logistically feasible to isolate the two organisms from each other during these experiments. As a result, the co-culture lipidome data reflects the composition of both organisms combined and our capacity to attribute shifts within it to either organism is limited. Despite this, the lipid profiles of *Ca*. Nha. antarcticus biomass pre- and post-incubation were comparable (Fig. 1e, f, *Ca*. Nha. antarcticus data point and error bars represent both samples) indicating most changes in co-culture lipids were likely host derived. However, it remains possible that during intermediate timepoints *Ca*. Nha. antarcticus is responsible for some of the variation in lipid abundance observed. Development of novel techniques for isolating *Ca*. Nha. antarcticus from *Hrr. lacusprofundi* in the future may provide greater clarity regarding the source of changes in the lipidomes of the two organisms in co-culture.

Our study revealed that the DPANN archaeon *Ca*. Nha. antarcticus selectively acquires specific lipids from its host *Hrr. lacusprofundi*. Additionally, during co-cultivation, *Hrr. lacusprofundi* modified its own lipid composition likely resulting in changes in membrane integrity that may constitute a lipid defense mechanism to restrict the exploitability of host cells by the symbiont. This study also emphasizes the strength of employing computational untargeted lipidomics approaches to elucidate lipid interactions within a host-symbiont culture system. Further research is needed to delve into the mechanisms underlying the specific selection of lipids and lipid defensive functions by the symbiont and

host, respectively. Understanding these mechanisms will deepen our knowledge of archaeal host-symbiont interactions at a molecular level, especially in the context of lipid exchange and survival strategies in extreme environments.

## Methods

### Purification of nanohaloarchaeal cells

To acquire *Ca*. Nha. antarcticus cells for use in co-culture experiments, nanohaloarchaeal cells were purified via filtration following previously described methods[26]. Briefly, 1 L of nanohaloarchaeal enrichment culture[9] was filtered sequentially through 0.8 μm (3x) and 0.2 μm (3x) polycarbonate filters (Isopore, Merck Scientific). Purified cells were then pelleted by centrifugation at 20,000 $g$ for 10 min and resuspended in 4 mL of DBCM2 media[55]. 100 μL of purified cells were stained with MitoTracker Green (as previously described[26]) and Nile Red (1 μg/ mL in 30% Salt Water (SW) mix[55]) for 1 h and imaged on an Axio Imager M2 microscope to assess purity and possible contamination of cell debris. 100 μL of purified cells were pelleted, DNA extracted using a PeqGold DNA Blood and Tissue Extraction kit following the manufacturer's instructions (VWR), and PCR performed targeting both *Ca*. Nha. antarcticus and *Hrr. lacusprofundi* (primer details in Table 1) to confirm purity of cells. Remaining *Ca*. Nha. antarcticus cells were divided into four aliquots (~5 × 10$^7$ cells): one aliquot was pelleted and used as biomass for lipidomic analyses, whilst the other three aliquots were inoculated into pure cultures of *Hrr. lacusprofundi* R1S1 for co-culture experiments.

### Cultivation experiments

Cultures of *Hrr. lacusprofundi* R1S1B were grown in DBCM2 in triplicate by shaking (120 r.p.m.) at 30 °C in volumes of 250 mL to late exponential phase. Cultures were then split into duplicates and diluted to an OD$_{600}$ of 0.2 in 250 mL. Infected cultures were inoculated with ~5 × 10$^7$ purified nanohaloarchaeal cells (see above). Samples for downstream analyses were harvested at 0 h, 6 h, 12 h, 24 h, and 48 h. Culture density was measured using OD$_{600}$ in triplicate with fresh DBCM2 media as blank at each timepoint. For lipidomics samples, 10 mL of culture was pelleted at 6000 $g$ for 30 min then washed in 30% SW and pelleted at 20,000 $g$ for 10 min three times to remove excess media components. Samples for qPCR were acquired by pelleting 1 mL of culture at 20,000 $g$, and removal of supernatant, and DNA extraction were performed as above. Samples for FISH were fixed with 2.5% glutaraldehyde at 4 °C overnight, washed with milliQ water and stored at −20 °C. In addition to *Hrr. lacusprofundi* R1S1 cultures and *Hrr. lacusprofundi* R1S1B – *Ca*. Nha. antarcticus co-cultures, lipidomics samples were collected from the nanohaloarchaeal enrichment culture, a pure *Natrinema* sp. isolated from the enrichment, and five additional and distinct *Hrr. lacusprofundi* strains as controls.

### PCR and qPCR

All PCR and qPCR reactions were performed using the same primer pairs for either *Ca*. Nha. antarcticus or *Hrr. lacusprofundi* (Table 1). Purity of filtered *Ca*. Nha. antarcticus cells was assessed by standard PCR with both sets of primers using DreamTaq polymerase (ThermoFisher) for 35 cycles at 55 °C annealing temperature. Standards for qPCR were

produced by PCR amplification of both *Ca*. Nha. antarcticus and *Hrr. lacusprofundi* 16S rRNA genes followed by cloning of amplified products into a pGEM-T easy vector (Promega), transformation of JM109 competent cells (Promega), and subsequent plasmid purification using a peqGOLD Plasmid Miniprep Kit, all steps were carried out as per the manufacturer's instructions. qPCR reactions were performed in a CFX96 Real-Time PCR Detection System (Bio-Rad) for 40 cycles with an annealing temperature of 55 °C.

## Fluorescence microscopy

Fluorescence in-situ Hybridization reactions were carried out as previously described[9,56]. Briefly, fixed samples were pelleted at 20,000 *g* for 10 min, resuspended in hybridization buffer with probes (100 pM working concentration, probe details in Table 1) specific to *Ca*. Nha. antarcticus and *Hrr. lacusprofundi* and incubated at 46 °C for 3 h. Cells were then pelleted at 20,000 *g* for 10 min, resuspended in wash buffer, and incubated at 48 °C for 30 min. Cells were then pelleted again at 20,000 g for 10 min, resuspended in PBS with 300 nM DAPI for counterstaining, and incubated for 1 h. Stained cells (30 µL) were mounted onto glass slides with antifadant and imaged on an Axio Imager M2. Data analysis and image processing was conducted using Fiji[57].

## Genomic analyses

To identify lipid biosynthesis capacities within *Ca*. Nha. antarcticus and *Hrr. lacusprofundi* genomes were annotated with a suite of functional annotation tools and manually curated. Due to rapid genome rearrangement within *Hrr. lacusprofundi* strains *Hrr. lacusprofundi* strain R1S1 was re-sequenced and assembled prior to conducting experiments. To avoid confusion with the existing *Hrr. lacusprofundi* R1S1 genome, we designate this as *Hrr. lacusprofundi* R1S1B. DNA was extracted as described above and sent for sequencing with Illumina 2 × 150 bp paired end reads (Eurofins). Raw reads were trimmed with trimmomatic[58] (v0.36, settings: SLIDINGWINDOW:5:22 MINLEN:100), assembly was performed using SPAdes[59] (v3.15.0, settings: -t 10 -k 21,39,59,99,127) and assembly was quality assessed with quast[60] (v4.6.3). Genomic analysis of *Ca*. Nha. antarcticus was carried out on the published genome accessed from IMG (IMG Genome ID: 2643221421). For both genomes the approach from ref. 6 was followed. Briefly, coding sequences were predicted using Prokka v1.14[61] (settings: --kingdom Archaea --addgenes --force --increment 10 --compliant --centre UU --cpus 20 --norrna −notrna). For functional annotation of genes several additional databases were used including COGs[62] (downloaded October 2020), arCOGs[63] (2018 version), KO profiles from the KEGG Automated Annotation Server[64] (downloaded April 2021), the Pfam database[65] (release 34.0), the TIGRFAM database[66] (release 15.0), the Carbohydrate-Active enZymes (CAZy) database[67] (v7, downloaded August 2020), the Transporter Classification Database[68] (downloaded April 2021), the Hydrogenase database[69] (HydDB, downloaded July 2020), and NCBI_nr (downloaded Aug 2021). In addition to this, protein domain predictions were carried out using InterProScan[70] (v5.62-94.0, setting: --iprlookup --goterms).

Annotations for the respective databases were carried out as follows. COGs, arCOGs, KOs, PFAMs, TIGRFAMs, and CAZymes were all identified using hmmsearch v3.1b2 (settings: -E 1e-5). The Transporter Classification Database and Hydrogenase Database were queried using BLASTp[71] v2.7.1 (settings: -evalue 1e-20). For database searches the best hit was selected based on highest e-value and bitscore and summarized in Supplementary Data 2 and 4. Multiple hits were allowed for InterProScan domain annotations using a custom script for parsing results (parse_IPRdomains_vs2_GO_2.py). Best blast hits against the NCBI_nr database were identified using DIAMOND[72] (settings: blastp --more-sensitive --evalue 1e-5 --no-self-hits). Identification of lipid biosynthesis genes was performed manually by screening annotated genes for KEGG annotations associated with synthesis of relevant lipids.

## Lipidome extraction and analysis

The methodology for lipid extraction and measurement in this study is detailed in ref. 29 To eliminate background lipids and contaminants, both medium blanks and extraction blanks were utilized. Briefly, the samples and blanks were extracted using a modified Bligh-Dyer extraction method[73,74]. They were subjected to ultrasonic extraction for 10 min, twice using a mixture of methanol, dichloromethane (DCM) and $[PO_4^{3-}]$ buffer (2:1:0.8, v/v/v) and twice with a mixture of methanol, DCM, and aqueous trichloroacetic acid solution at pH 3 in the same ratio. The organic phase was separated by adjusting the solvent mixture to a final ratio of 1:1:0.9 (v/v/v) with additional DCM and buffer. This organic phase was then subjected to three further extractions using DCM and then dried under a stream of $N_2$ gas. The dry extract was re-dissolved in a methanol and DCM mixture (9:1, v/v), followed by filtration through 0.45 µm regenerated cellulose syringe filters (4 mm diameter; Grace Alltech). The filtered extracts were subsequently analyzed using an Agilent 1290 Infinity I UHPLC system coupled to a Q Exactive Orbitrap MS (Thermo Fisher Scientific, Waltham, MA). The generated output data from the UHPLC-HRMS² analysis were processed with MZmine software[75] to extract MS¹ and MS² spectra as well as quantify peaks. This processing included several steps: mass peak detection, chromatogram building, deconvolution, isotope grouping, feature alignment, and gap filling (https://ccms-ucsd.github.io/GNPSDocumentation).

## Molecular networking

The MS/MS spectra dataset was further processed using the Feature-Based Molecular Networking tool on the GNPS platform[76,77]. Molecular networking is a key data analysis methodology in untargeted metabolomics studies based on MS/MS analysis, arranges MS/MS spectra into a network-like map. In this map, molecules with similar spectral patterns are clustered together, indicating their structural similarities. The analysis involves calculating vector similarities comparing pairs of spectra based on at least five matching fragment ions (peaks). This comparison not only considers the relative intensities of the fragment ions but also the difference in precursor *m/z* values between the spectra[77,78]. The molecular network is constructed using MATLAB scripts, where each spectrum is linked to its top K scoring matches, usually allowing up to 10 connections per node. Connections (edges) between spectra are retained if they rank among the top K matches for both spectra and if the vector similarity score surpasses a predetermined threshold. The similarity score is quantified as a cosine value, where a score of 1.0 signifies identical spectra. In this study, a cosine value of 0.5 was used to define significant spectral similarities, indicating a moderate to high level of structural resemblance between the analyzed molecules.

In the molecular networking analysis of the MS/MS spectra, when an ion component displayed both protonated $[M + H]^+$ and ammoniated $[M + NH_4]^+$ ions, the overall abundance of that component was calculated as the combined total of the abundances of these two ion forms. For the construction of the molecular network, a minimum of five shared fragment ions was established as the criterion for connecting pairs of related MS/MS spectra with an edge. Each node within the network was permitted to connect to a maximum of ten analogs. In addition, consensus spectra were compared against the GNPS spectral library[77,79], allowing for a maximum analog mass difference of *m/z* 500. The maximum size of nodes allowed in a single connected subnetwork was capped at 100. In scenarios where the dataset contained a significant number of related lipids (exceeding 100), these lipids were segregated into different subnetworks.

The molecular networks derived from the analysis were visualized using Cytoscape version 3.9.1[80,81]. It is important to note that since many of the lipids detected in this study have not been previously characterized, authentic standards for absolute quantification were not available. The lipids were corrected for sample recovery with a 1,2-dipalmitoyl-sn-glycero−3-O-4'-[N,N,N-trimethyl(d9)]-homoserine

(DGTS-d9) internal standard then examined based on their normalized peak area responses. Consequently, the relative peak areas calculated abundance do not indicate the actual relative abundance of different lipids in samples. Nevertheless, this method allowed comparison of lipids between different cultures or cultivation conditions, rather than determining the absolute quantities of each lipid present[82].

## Information theory framework

The lipidome's diversity and specialization, along with the specificity of individual lipid species, were defined and analyzed using an information theory framework[38,39,83]. Lipids were characterized via their distinct tandem $MS^2$ spectra and their relative occurrence frequencies across various cultures. The lipidome diversity, the $H_j$ index, was calculated using the Shannon entropy based on the frequency distribution of lipid species as determined by the abundance of their $MS^2$ precursor ions. The equation is as follows

$$H_j = -\sum_{i=1}^{m} P_{ij}\log_2(P_{ij}) \tag{1}$$

where $P_{ij}$ correspond to the relative frequency of the $i$th $MS^2$ ($i = 1, 2, ..., m$) in the $j$th sample ($j = 1, 2, ..., t$), to illustrate how abundant a specific $MS^2$ spectrum is relative to all others.

The average frequency of the $i$th $MS^2$ among samples was calculated as

$$P_i = \frac{1}{t}\sum_{j=1}^{m} P_{ij} \tag{2}$$

Individual lipid species specificity, the $S_i$ index, was defined as the identity of a given $MS^2$ regarding frequencies among all the cultures. The lipid species specificity was calculated as

$$S_i = \frac{1}{t}\left(\sum_{j=1}^{t} \frac{P_{ij}}{P_i}\log_2\frac{P_{ij}}{P_i}\right) \tag{3}$$

Individual lipid species specificity of specific cultures, was defined as $S_{ij}$ index.

$$S_{ij} = \sum_{j=1}^{t} \frac{P_{ij}}{P_i}\log_2\frac{P_{ij}}{P_i} \tag{4}$$

The lipidome specialization $\delta_j$ index was measured as the average of the $MS^2$ specificities using the following formula

$$\delta_j = \sum_{i=1}^{m} P_{ij}S_i \tag{5}$$

## Statistical analysis

For PCA, the abundance data of lipid species were initially transformed using the Hellinger distance method[84] to mitigate bias arising from zero values. This data was then processed and visualized using R software, version 4.1.2. Hierarchical clustering was performed using the "ggplot2" and "pheatmap" packages in R, version 4.3.2.

## Reporting summary

Further information on research design is available in the Nature Portfolio Reporting Summary linked to this article.

## Data availability

Source data are provided with this paper. The processed data (.mgf and.csv) with the molecular network and detailed parameter settings used in this study are available at the GNPS platform under accession code https://gnps.ucsd.edu/ProteoSAFe/status.jsp?task= c75ddcdd8d2e426e9d537ee1037a2b43. The raw data of lipidome used in this study is available in MassIVE under accession code https://massive.ucsd.edu/ProteoSAFe/dataset.jsp?accession=MSV000094377. The *Halorubrum lacusprofundi R1S1B* genome used in this study has been deposited at DDBJ/ENA/GenBank under the accession JAXGGM000000000, BioProject: PRJNA1046704, BioSample: SAMN38507334. The version described in this paper is version JAXGGM010000000. Databases used for genome annotation can be found at: COGs 2020 update https://ftp.ncbi.nlm.nih.gov/pub/COG/COG2020/data/, arCOGs 2018 https://ftp.ncbi.nlm.nih.gov/pub/wolf/COGs/arCOG/ KEGG 2021 https://www.kegg.jp/kegg/download/, TIGR-FAMs 15.0 https://ftp.ncbi.nlm.nih.gov/hmm/TIGRFAMs/release_15.0/, CAZy Database 7 http://www.cazy.org/spip.php?rubrique59, Hydrogenase Database 2020 https://services.birc.au.dk/hyddb/browser/, NCBI_nr 2021 https://ftp.ncbi.nlm.nih.gov/blast/db/, Transporter Classification Database 2021 https://www.tcdb.org/download.php. The source data for all the figures are provided either in the Supplementary information or in the extended dataset in our data repository on zenodo https://doi.org/10.5281/zenodo.10851289.

## Code availability

All custom scripts and workflows used to generate data in this study are available in zenodo under accession code https://doi.org/10.5281/zenodo.10851289.

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

## Acknowledgements

We thank Michel Koenen for the lipidome extraction. We acknowledge Wen-Cong Huang and Dina Castillo Boukhchtaber for their support in the culturing experiment and Prof. Stefan Schouten, Prof. Laura Villanueva, Dr. Kerstin Fiege, and Dr. Diana Sahonero Canavesi who provided valuable comments on the initial experiment. J.S.S.D. received funding from the European Research Council (ERC) under the European Union's Horizon 2020 research and innovation program (grant agreement no.694569—MICROLIPIDS) and from a Spinoza award from NWO. A.S. has received funding from the European Research Council (ERC) under the European Union's Horizon 2020 research and innovation programme (grant agreement No. 947317, ASymbEL), the Moore–Simons Project on the Origin of the Eukaryotic Cell, Simons Foundation 735929LPI, and a Gordon and Betty Moore Foundation's Symbiosis in Aquatic Systems Initiative (GBMF9346).

## Author contributions

S.D., J.N.H., N.J.B., J.S.S.D., and A.S. conceived the study. J.N.H. performed all cultivation and microscopy analyses. S.D. performed lipidome data analyses. S.D. and N.J.B. performed lipid identifications. S.D. and J.N.H. wrote the draft manuscript. J.S.S.D. and A.S. supervised the study. All authors read, discussed, and approved the final version of the manuscript.

## Competing interests

The authors declare no competing interests.
