## [Peer Review File · Nature Communications]

Selective lipid recruitment by an archaeal DPANN symbiont from its hostREVIEWER COMMENTS

Reviewer #1 (Remarks to the Author):

The paper by Ding describes the selective enrichment of lipids by *Candidatus Nanohaloarchaeum antarcticus* from its host *Halorubrum lacusprofundi*. This result seemingly contrasts with previous experiments with *Nanoarchaeum equitans* and *Ca. Micrarchaeum harzensis* and is the first demonstration of selective lipid uptake by an archaeal symbiont. This paper is exceptionally focused and was well-conceived, well-executed, and well-described. The design to set up temporary pure co-cultures in order to control the experiments is outstanding. The results, figures, and discussion are exceptionally clear. The methods are robust the work is done by experts. To the best of my knowledge, all conclusions are well-supported and justified. I congratulate the authors and thank them for their excellent work and very clear analysis and writing. I have only a few minor suggestions that I hope improve the paper.

Major or substantive criticisms:

- None

Minor points, questions, and suggestions:

- If possible, please write out the lipid names and the acronyms in Figure 1d and/or 2a so the lipid groups can be more easily understood for non-lipid specialists.
- Line 73: Are the core lipids functional? Or are they biosynthetic or degradation intermediates? Maybe it's worth clarification for non-experts on lipids.
- Line 95: Comma before which and after strains
- Figure 1 e, f, g. Please describe the acronyms on the plots better. I assume Cc_0 is the co-culture at 0 hours, but this isn't totally clear. Please indicate *Natrinema* in the key.
- Line 112: subject missing = us (if you don't mind first person)
- Line 113 and Figure 1g, please number the clusters on the figure and list the clusters with numbers in the text to help connect the text to the figure.
- Line 120: "the same as that of" – consider "as were lipid profiles"
- Line 143: varied. Comma after samples.
- Line 147. I hope this section and Figure 3 can be modified so it better explains the microbiology. Figure 3: Set size = number of lipids? Genre???? Intersection size?? Please either modify these axis labels so they are understandable (e.g., number of lipids) or make a better effort to explain the results in the context of microbiology. In line 148, what is a reader supposed to understand regarding the text: "interrelations between study objects"? What is "study objects" in line 156? Please clarify this section.
- Line 159-160, menaquinones is misspelled. I'm sensitive to spelling errors and this is all I found (nice job!), but please check spelling throughout.
- Line 182 = archaea
- Line 190, phosphatidylcholine-based (this is a compound adjective, so a hyphen is recommended for clarity)
- Line 192, use a comma, not a semicolon, because the text before the semicolon can't stand alone as a sentence.

Reviewed by Brian Hedlund

Reviewer #2 (Remarks to the Author):

The Ding et al. paper uses a DPANN symbiont-host system to demonstrate the power of lipidomics in examining the DPANN-host interactions. The data showed that the symbiont *Ca. Nha. antarcticus* selectively obtained membrane lipids from the host *Hrr. Lacusprofundi*. This observation is in contrast to two previous publications that claimed non-selective lipid acquisitions by the symbiont from its host (Jahn et al., 2004 "Composition of the lipids of *Nanoarchaeum equitans* and their origin from its host *Ignicoccus* sp. strain KIN4/I" and Krause et al. 2022 "The importance of biofilm formation for cultivation of a *Micrarchaeon* and its interactions with its *Thermoplasmatales* host") Ding et al. speculated that this discrepancy may be due to

"natural differences between these distantly related symbiotic partnerships or could stem from differences in the methodologies." My first reaction would be that the method problem should be resolved immediately, which should be easily done by performing the same lipidomic procedure on the one or two published DPANN symbiont-host systems (i.e. Nanoarchaeum equitans and its host by Jahn et al. (2004) and Micrarchaeon and its host by Krause et al. (2022)). It is very important to determine whether all three DPANN symbiont-host systems give consistent results that the symbiont selectively acquires lipids from its host, albeit among different phyla (e.g., Thermoproteota, Thermoplasmata, Halobacteria) and distinct membrane structures (e.g., bilayer or monolayer cell membrane) of the hosts. If yes, it means that the DPANN symbiont is "smart" and can orchestrate its interaction with the host. It is equally important to know that some symbionts non-selectively obtain lipids from their host and how the symbionts employ the diverse lipid acquisition strategies. I would support the publication of this paper in NC and would encourage the authors to do the same lipidomic analysis on the other symbiont-host systems.

Major comment:

Despite the significant interest this study may inspire; my main concern is the implication underlying the observation. The authors have made a good effort in interpreting their data and trying to discuss why the symbiont selectively acquires specific lipids from its host. The specific selection of lipids appears to be related to the high divalent cation environment and the metabolic efficiency but the mechanisms are lacking. Is this selection driven by environmental stress or genetic preference? Also, could it be a common phenomenon in the DPANN symbiont-host systems or just specific to this system? Although these questions cannot be fully resolved, the authors could combine transcriptomics and lipidomics analyses to provide some insights into the physiological mechanisms underlying the observation of lipid selection based on more experimental data.

Line 16: remove "that there is"

Line 21-23: This sentence is confusing. You mean a stable co-culture of Hrr. Lacusprofundi and Ca. Nha. Antarcticus must be maintained in an enrichment culture with other strains. How about the procedure shown in Fig. 1a? The Nha cells are separated from the enrichment by filtering and then mixed only with HP cells to form co-culture? This co-culture seems to be stable without other strains?

Line 42: The lipidomic data of the QC samples were absent in Figure 1 and Figure 3. Please provide the names and abbreviations of 6 strains for the QCs.

Line 77: Here, SDG stands for sulphated diglycosyl, but it is not used in the following text or figure.

Line 158: "...between 24 and 40 h." You mean 24 and 48 h?

Line 404 (Figure Caption 1): Here "S" is used as the abbreviation of PGS. However, in your figures, "S" comes with G/2G, and coexist with PGS. Is this correct?

Line 222-223: The function of MKs and bacterioruberins should be introduced.

Line 227-228: "As mentioned above, the reduced CL abundance within the membrane may act to increase membrane stability and resistance to mechanical stress." The sentence could lead to misunderstanding; please rephrase it.

Line 234-236: The sentence indicates the host actively modulates its membrane to defend the symbiont. However, no data show that such modulates help resist the invasion.

Figure 1d: The title of the y-axis (Lipid Classes) is smaller than that of the x-axis.

Figure 2: It is quite challenging to recognize the changes in lipid profiles of both the host and the co-culture samples collected at different time points in this figure. Please emphasize the remarkable changes between the host and the co-culture at each time point or with the time series, making it easier to capture the vital information in this figure.

Figure 3b: Does the second column of the dot-matrix plot represent the lipid species (with the intersection size of 30) shared by none of the samples? And this value turns into 14 in Fig. S8 when the enrichment sample (CLAC2B) is added. The CLAC2B sample has 16 unique lipid species according to Fig. S8? Please clarify this on Lines 160-164 and explain why the CLAC2B sample that should be the initial condition before the co-culture was excluded in Fig. 3?

The expression of lipids is not consistent. For example, a group of lipids in legend is shown as "PG". while in other places, some expressions were "PG-AR", "Unknown PG", etc. Figures should also be cited in the Discussion chapter.

Reviewer #3 (Remarks to the Author):

The paper by Ding et al. provides a detailed lipidomics analysis of a DPANN symbiont-host system that improves our understanding of symbiont host interactions and the role lipids may play in it. The authors report interesting observations and provide plausible explanations for them. The fact that they observe differences in lipid composition that earlier analyses of DPANN symbiont-host systems have not, is surely in large part due to less sensitive detection techniques that were used back in those days. In addition, many of the currently investigated molecules where major changes were observed in this manuscript (such as the quinones or carotenoids) were not even part of these earlier studies.

I have some questions about the visualization of the data that is described in the text. For instance, in line 95 the authors write that Fig. 1e (and the methods) show the proximity of *Natrinema* sp. to the host in terms of lipid composition. I don't see this in Fig. 1e, I also don't see this in any of the PCA plots in Fig. 1e or 1f. Which colored dot is the *Natrinema* sp. sample? It is not explained in the figure caption or the methods what the abbreviations for the yellow dots stand for. My searches for the presentation of the lipid results for *Ca. Nha. Antarcticus* post co-culture experiment are also in vain or I simply don't know where to look for them. Where can I find these in Fig. 1?

In general, I would have found it interesting to look at the co-culture experiment separately into lipid compositional changes of the host and the symbiont. If I understand it correctly in Figs. 1g, 2 and 3 the authors only show the lipid results of the entire co-culture (symbiont + host). This way their discussion on lipid changes become more speculative because they cannot say to which part the lipid composition is driven by the host vs the symbiont.

I wonder about the utility of Fig. 2 to bring the story across. While it is a very pretty figure, I have a hard time assessing the visually only subtle differences in these network plots. Furthermore, as most of the discussion focusses on Fig. 3 this figure does not even seem necessary to me. I would therefore recommend to remove Fig. 2 or move it to the supplement. Similarly, I don't see the necessity to show Figs. 1b and 1c in the main text. Fig. 1 is as it is already jam-packed with information.

Specific comments.

Line 160: Change to "menaquinones"

Line 354: I would use the word "normalized" instead of "calibrated" since it sounds misleading to write that "peak area responses were calibrated" when in fact they were not.

Line 354: I don't understand the use of the word "necessarily". The standard DGTS for normalization has structurally very little in common with the investigated lipid compounds, therefore, there is no reason to assume that the use of this standard may have accounted for differences in peak area responses. Thus, I suggest to remove the word "necessarily".

Line 438: The 'S' should be in upper-case for 16S rRNA

Fig. 1 I find the structure of the figure panels slightly confusing. A, b and c are vertically aligned, but d, e, f and g are horizontally aligned? Is there a purpose for this disorder? I would also advise the authors to not only rely on color when making their plots. It's always helpful to use different symbol types to be considerate of color blind people.

Fig. 2: This figure looks fantastic, but I don't see a lot of additional gain by showing this (see comments above).

Fig. 3a: What do the letters a,b,c,d stand for above the individual bars?

Fig. 3b: I don't understand the x-axis in this plot. What is meant by genre? Please explain this in the figure caption. What do grey dots mean?

REVIEWER COMMENTS

Reviewer #1 (Remarks to the Author):

The paper by Ding describes the selective enrichment of lipids by Candidatus Nanohaloarchaeum antarcticus from its host Halorubrum lacusprofundi. This result seemingly contrasts with previous experiments with Nanoarchaeum equitans and Ca. Micrarchaeum harzensis and is the first demonstration of selective lipid uptake by an archaeal symbiont. This paper is exceptionally focused and was well-conceived, well-executed, and well-described. The design to set up temporary pure co-cultures in order to control the experiments is outstanding. The results, figures, and discussion are exceptionally clear. The methods are robust the work is done by experts. To the best of my knowledge, all conclusions are well-supported and justified. I congratulate the authors and thank them for their excellent work and very clear analysis and writing. I have only a few minor suggestions that I hope improve the paper.

Authors: We greatly appreciate the extremely positive comments by the reviewer.

Major or substantive criticisms:

- None

Minor points, questions, and suggestions:

- If possible, please write out the lipid names and the acronyms in Figure 1d and/or 2a so the lipid groups can be more easily understood for non-lipid specialists.

Authors: Due to the limited space of figures, we are not able to write the full names of lipids in the figure, however, the full names of lipids are given in the caption of Figure 1.

- Line 73: Are the core lipids functional? Or are they biosynthetic or degradation intermediates? Maybe it's worth clarification for non-experts on lipids.

Authors: We do not know what their function is, but we see them more often in archaeal lipid extracts and they could be biosynthetic or degradation products (either during extraction or in the culture itself). They are very recalcitrant and last for millions of years in the geological record so if lipids lose their polar headgroups the core lipids likely would not break down. We added a short sentence to clarify this.

- Line 95: Comma before which and after strains

Authors: Corrected

- Figure 1 e, f, g. Please describe the acronyms on the plots better. I assume Cc_0 is the co-culture at 0 hours, but this isn't totally clear. Please indicate Natrinema in the key.

Authors: This is correct, we have clarified the sample names in the caption of Figure 1 and given the Natrinema its own colour to highlight it. In line 410-412, we wrote 'Sample abbreviations: Ca. Nha. antarcticus (Nha), Hrr. lacusprofundi (HP), co-cultures (Cc), Natrinema sp. (NATC283), Enrichment

(CLAC2B), other *Hrr. Lacusprofundi* strains (DL11, DL14, DL12MDS, R1A8, ACAM34). The sample abbreviation with a number stands for its culturing time, for example, CC_6 means co-cultures sampled at 6 h.'

- Line 112: *subject missing = us (if you don't mind first person)*

Authors: Corrected

- Line 113 and Figure 1g, *please number the clusters on the figure and list the clusters with numbers in the text to help connect the text to the figure.*

Authors: We added this information in both the text and the figure 1g as the reviewer suggested.

- Line 120: *“the same as that of” – consider “as were lipid profiles”*

Authors: We have made the suggested edit.

- Line 143: *varied. Comma after samples.*

Authors: Corrected

- Line 147. *I hope this section and Figure 3 can be modified so it better explains the microbiology. Figure 3: Set size = number of lipids? Genre???? Intersection size?? Please either modify these axis labels so they are understandable (e.g., number of lipids) or make a better effort to explain the results in the context of microbiology. In line 148, what is a reader supposed to understand regarding the text: “interrelations between study objects”? What is “study objects” in line 156? Please clarify this section.*

Authors: We have made the changes as the reviewer suggested. In Figure 3, we have changed “set size” and “Genre” as “number of lipid species”, “Intersection size” as “number of lipids shared”. We have also modified the text accordingly to make it easier for the audience to understand.

- Line 159-160, *menaquinones is misspelled. I'm sensitive to spelling errors and this is all I found (nice job!), but please check spelling throughout.*

Authors: Corrected

- Line 182 = *archaea*

Authors: Corrected

- Line 190, *phosphatidylcholine-based (this is a compound adjective, so a hyphen is recommended for clarity)*

Authors: Corrected

- Line 192, use a comma, not a semicolon, because the text before the semicolon can't stand alone as a sentence.

Authors: Corrected

Reviewed by Brian Hedlund

Reviewer #2 (Remarks to the Author):

*The Ding et al. paper uses a DPANN symbiont-host system to demonstrate the power of lipidomics in examining the DPANN-host interactions. The data showed that the symbiont *Ca. Nha. antarcticus* selectively obtained membrane lipids from the host *Hrr. Lacusprofundi*. This observation is in contrast to two previous publications that claimed non-selective lipid acquisitions by the symbiont from its host (Jahn et al., 2004 “Composition of the lipids of *Nanoarchaeum equitans* and their origin from its host *Ignicoccus* sp. strain KIN4/I” and Krause et al. 2022 “The importance of biofilm formation for cultivation of a *Micrarchaeon* and its interactions with its *Thermoplasmatales* host”) Ding et al. speculated that this discrepancy may be due to “natural differences between these distantly related symbiotic partnerships or could stem from differences in the methodologies.” My first reaction would be that the method problem should be resolved immediately, which should be easily done by performing the same lipidomic procedure on the one or two published DPANN symbiont-host systems (i.e. *Nanoarchaeum equitans* and its host by Jahn et al. (2004) and *Micrarchaeon* and its host by Krause et al. (2022)). It is very important to determine whether all three DPANN symbiont-host systems give consistent results that the symbiont selectively acquires lipids from its host, albeit among different phyla (e.g., *Thermoproteota*, *Thermoplasmata*, *Halobacteria*) and distinct membrane structures (e.g., bilayer or monolayer cell membrane) of the hosts. If yes, it means that the DPANN symbiont is “smart” and can orchestrate its interaction with the host. It is equally important to know that some symbionts non-selectively obtain lipids from their host and how the symbionts employ the diverse lipid acquisition strategies. I would support the publication of this paper in NC and would encourage the authors to do the same lipidomic analysis on the other symbiont-host systems.*

Authors: We thank the reviewer for overall positive feedback. We agree that application of our lipidomics analyses to other DPANN - host systems would be beneficial for determining whether the differences between our results and those previously published are due to differences in biology or technique. However, we feel that it is not our task to check all former results reported in the literature with our methods. Specifically, such efforts would be extremely time intensive and represent an entirely new study suitable for a prospective project rather than an addition to this work. We have aimed at putting our study into the context of current literature but do think that re-examining the previous literature experimentally is beyond the scope of this work. Cultivation of DPANN archaea is extremely challenging and performing these experiments would require extensive additional work and financial support over the course of many months.

The two other DPANN symbiont host systems for which lipidomics data exists are *Ca. M. harzensis* and its host *Scheffleriplasma hospitalis*, as well as *N. equitans* and its host *I. hospitalis*. The former are extreme acidophiles that grow at a pH of 2 and only achieve cell densities of $\sim 10^7$ cells/mL, an order of magnitude lower than our chosen system (Krause et al. 2022, <https://doi.org/10.1038/s41467-022->

29263-y). The growth rates of these organisms are incredibly slow, and cultures take ~55 days to reach maximum cell density. Assuming a technique for purification could be developed, acquiring the necessary *Ca. M. harzensis* biomass to replicate our approach would require 2 L of culture per replicate (6 L total), grown in anaerobic conditions, in extremely acidic media, over a period of 2 months (excluding the time necessary to scale cultures up to this volume). In addition to this, the host organism is extremely sensitive to physical stresses because it lacks a cell wall and simply agitating the cultures during growth results in lysis. As a result, harvesting biomass of the symbiont through filtration, as we did for *Ca. Nha. antarcticus*, is not feasible and would require methods development.

N. equitans and *I. hospitalis*, the second system, represent extreme thermophiles and piezophiles that grow at ~95°C and 1.4 BAR under a H₂/CO₂ atmosphere (Huber et al. 2002, <https://doi.org/10.1038/417063a>). To get sufficient material for lipidomic and transcriptomic analyses they are typically cultivated in large bioreactors. Through correspondence with the group that performed the original lipidomics on these organisms we know that the production of purified *N. equitans* cells for lipidomics required a 300 L bioreactor incubation with constant passaging of fresh H₂ gas into the headspace of the reactor (this is the only technique that produces large amounts of unattached symbiont cells otherwise *N. equitans* is only found attached to the host). This cultivation process produced sufficient *N. equitans* biomass for a single lipidomics sample. We do not currently have access to the facilities necessary to perform such large-scale incubations to apply our technique to the *N. equitans* system. Therefore, performing these experiments as proposed would be extremely challenging. Therefore, we hope that the referee agrees that the suggested additional experiments are beyond the scope of the current submission.

Major comment:

Despite the significant interest this study may inspire; my main concern is the implication underlying the observation. The authors have made a good effort in interpreting their data and trying to discuss why the symbiont selectively acquires specific lipids from its host. The specific selection of lipids appears to be related to the high divalent cation environment and the metabolic efficiency but the mechanisms are lacking. Is this selection driven by environmental stress or genetic preference? Also, could it be a common phenomenon in the DPANN symbiont-host systems or just specific to this system? Although these questions cannot be fully resolved, the authors could combine transcriptomics and lipidomics analyses to provide some insights into the physiological mechanisms underlying the observation of lipid selection based on more experimental data.

Authors: We agree with the reviewer that developing a mechanistic understanding of the processes that underlie this selection would be beneficial and this is something we plan to explore in future work. However, it is not clear to us how the application of transcriptomics would accomplish this. It has been previously shown in other DPANN systems that the presence of DPANN symbionts result in significant shifts in host gene expression across multiple aspects of the host physiology (Giannone et al. 2014, doi: 10.1038/ismej.2014.112). During our analysis of the *Ca. Nha. antarcticus* genome we did not identify genes that stood out as particularly likely to play a role in scavenging lipids from the host. It is likely that any genes responsible for this belong to the large number of hypothetical proteins encoded by the symbiont's genome and transcriptomics would not resolve this because the nanohaloarchaeon requires its host for much more than just lipids and it is likely that much of the genome would exhibit differences in expression during the co-cultivation experiments.

Addressing the question of mechanism would require the identification and functional characterization of putative genes coding for proteins involved in lipid transfer, after which we could then explore expression during co-cultivation. This is possible but the amount of work involved is extensive and it would require over a year of additional experimentation and an additional research grant supporting the

costs arising from this. So whilst we are in agreement that it would be useful, in our opinion it is beyond the scope of the current work.

Line 16: remove “that there is”

Authors: Corrected

Line 21-23: This sentence is confusing. You mean a stable co-culture of Hrr. Lacusprofundi and Ca. Nha. Antarcticus must be maintained in an enrichment culture with other strains. How about the procedure shown in Fig. 1a? The Nha cells are separated from the enrichment by filtering and then mixed only with HP cells to form co-culture? This co-culture seems to be stable without other strains?

Authors: The culture is unstable in the sense that eventually the parasitic behavior of the nanohaloarchaeon results in either destabilization of the host population or loss of the nanohaloarchaeon (presumably through rejection of the parasite by the host). This method used for producing co-cultures has been used multiple times (see Hamm et al. 2019 <https://doi.org/10.1073/pnas.1905179116>, and Hamm et al. 2023 <https://doi.org/10.1101/2023.02.24.529834>) and long-term instability reported in the original paper describing the system (Hamm et. al. 2019). Over the time period our experiments were conducted, the stability is not an issue, which is partly why we opted for a short time frame. We have added a short statement to clarify this in the introduction.

Line 42: The lipidomic data of the QC samples were absent in Figure 1 and Figure 3. Please provide the names and abbreviations of 6 strains for the QCs.

Authors: Lipidomic data for the controls are included in Fig. 1e and f, represented by yellow dots. We have refined the sample descriptions to make this more clear. The purpose of these controls was not to serve as controls for the time series per se, but rather to ensure that strain variations did not influence the host lipid composition by serving as controls for the initial culture conditions. The fact that all controls clustered with the 0-hour timepoint from the pure culture of the strain used in the time-series experiments in the PCA and information theory analysis (Fig. 1e and f) reassured that strain variation did not pose a concern. Additionally, we have now included controls in Fig. 3a and supplementary Fig. S8 and S9, using the average abundances of *Natrinema* sp. (NATC283) and five other *Hrr. Lacusprofundi* strains (DL11, DL14, DL12MDS, R1A8, ACAM34). We hope this provides a satisfactory explanation.

Line 77: Here, SDG stands for sulphated diglycosyl, but it is not used in the following text or figure.

Authors: We thank the reviewer for pointing this out. We’ve made the change in the text accordingly “as well as non-phospholipids, including sulfur containing lipids except PGS (S, e.g., sulphated diglycosyl 3 species), monoglycosyl (1G, 2 species), diglycosyl (2G; 2 species) archaeol.”

Line 158: “...between 24 and 40 h.” You mean 24 and 48 h?

Authors: Yes, this has been corrected.

Line 404 (Figure Caption 1): Here “S” is used as the abbreviation of PGS. However, in your figures, “S” comes with G/2G, and coexist with PGS. Is this correct?

Authors: In figure caption 1, “S” was used as the abbreviation of archaeol lipids containing a sulfur-containing head group except for PGS, for example, sulphated diglycosyl and many other unknown sulfur lipids. We have rephrased this sentence for clarification.

Line 222-223: The function of MKs and bacterioruberins should be introduced.

Authors: The function of bacterioruberins was introduced in the previous paragraph discussing absence of these lipids from *Ca. Nha. antarcticus*. We have added a brief statement introducing to menaquinones to this paragraph as well (as this paragraph also contains discussion surrounding menaquinones).

Line 227-228: “As mentioned above, the reduced CL abundance within the membrane may act to increase membrane stability and resistance to mechanical stress.” The sentence could lead to misunderstanding; please rephrase it.

Authors: We have rephrased this sentence for clarification.

Line 234-236: The sentence indicates the host actively modulates its membrane to defend the symbiont. However, no data show that such modulates help resist the invasion.

Authors: We agree with the reviewer there is no data to show the impact of the changes in lipid composition on interactions with *Ca. Nha. antarcticus*. We have rephrased this sentence to better clarify our hypothesis regarding the impact this may have on the interactions. Indeed, we do not believe this is related directly to invasion but more likely to access to the host cytoplasm which is thought to precede invasion. Whilst our data does not directly confirm this, we believe the statement as currently written is a logical conclusion drawn from current understanding of how CL affects membrane stability and the context in which the change takes place.

Figure 1d: The title of the y-axis (Lipid Classes) is smaller than that of the x-axis.

Authors: Corrected

Figure 2: It is quite challenging to recognize the changes in lipid profiles of both the host and the co-culture samples collected at different time points in this figure. Please emphasize the remarkable changes between the host and the co-culture at each time point or with the time series, making it easier to capture the vital information in this figure.

Authors: We’ve put Fig. 2B-C into the supplementary as Rev#3 suggested. In addition, we added arrows with text to emphasize the remarkable changes between the host and the co-cultures as the reviewer suggested.

Figure 3b: Does the second column of the dot-matrix plot represent the lipid species (with the intersection size of 30) shared by none of the samples? And this value turns into 14 in Fig. S8 when the enrichment sample (CLAC2B) is added. The CLAC2B sample has 16 unique lipid species according to

Fig. S8? Please clarify this on Lines 160-164 and explain why the CLAC2B sample that should be the initial condition before the co-culture was excluded in Fig. 3?

Authors: Yes, this is a correct interpretation. The pool of lipid species used to produce this figure includes those found in control samples which is why some are not present in the samples shown in the figure. In our opinion the enrichment culture CLAC2B is not an initial condition because it is a complex mix of multiple host strains, the nanohaloarchaeon, and another haloarchaeal species, that is not at a comparable growth stage as the experimental samples. We opted against including it in the main text figure because the focus of the figure was on the differences between experimental samples, and we felt inclusion of this sample made it less straightforward to compare these. However, we added it for Fig. S9 because we recognized some readers may be interested in the overlap between this sample and the co-cultures / pure nanohaloarchaeal sample. We hope this clarifies our reasoning behind not including the enrichment in the main text figure. We have also added control samples as the reviewer suggested, *Natrinema* sp. (NATC283) and five other *Hrr. Lacusprofundi* strains (DL11, DL14, DL12MDS, R1A8, ACAM34).

The expression of lipids is not consistent. For example, a group of lipids in legend is shown as “PG” while in other places, some expressions were “PG-AR”, “Unknown PG”, etc.

Authors: We use the terms PG, PG-AR, and unknown PG to denote different lipid expressions because they represent distinct entities. PG-AR signifies a PG headgroup linked to an archaeol core lipid, characterized by a lipid species with an m/z 807.684 and an elemental composition of $C_{46}H_{96}O_8P^+$ (refer to the caption of Fig. 3a for details). The term PG encompasses all PG lipid species, including PG-AR and PG-EXT-AR (which has an additional isoprenoid chain), and others. The term 'unknown PG' is used when the core lipid is unidentified. We have now added a sentence to explain this in the figure caption of Fig. 3.

Figures should also be cited in the Discussion chapter.

Authors: We've cited the figures in the Discussion chapter as suggested.

Reviewer #3 (Remarks to the Author):

The paper by Ding et al. provides a detailed lipidomics analysis of a DPANN symbiont-host system that improves our understanding of symbiont host interactions and the role lipids may play in it. The authors report interesting observations and provide plausible explanations for them.

The fact that they observe differences in lipid composition that earlier analyses of DPANN symbiont-host systems have not, is surely in large part due to less sensitive detection techniques that were used back in those days. In addition, many of the currently investigated molecules where major changes were observed in this manuscript (such as the quinones or carotenoids) were not even part of these earlier studies.

*I have some questions about the visualization of the data that is described in the text. For instance, in line 95 the authors write that Fig. 1e (and the methods) show the proximity of *Natrinema* sp. to the host in terms of lipid composition. I don't see this in Fig. 1e, I also don't see this in any of the PCA plots in Fig. 1e or 1f. Which colored dot is the *Natrinema* sp. sample? It is not explained in the figure caption or the methods what the abbreviations for the yellow dots stand for. My searches for the presentation*

of the lipid results for Ca. Nha. Antarcticus post co-culture experiment are also in vain or I simply don't know where to look for them. Where can I find these in Fig. 1?

The Natrinema sample was originally included with the same colour as the additional host strains (NAT2BC, yellow) we have corrected this and given the Natrinema a unique colouration in the figures. We have also clarified the naming of the additional host strains to make this more clear. The post-cultivation nanohaloarchaeal sample (Nha_b) is clustered with the pre-cultivation sample (Nha_a) in the PCA and the two were pooled for statistical analyses in other figures because there were not major shifts in the nanohaloarchaeal lipidome during the incubation.

In general, I would have found it interesting to look at the co-culture experiment separately into lipid compositional changes of the host and the symbiont. If I understand it correctly in Figs. 1g, 2 and 3 the authors only show the lipid results of the entire co-culture (symbiont + host). This way their discussion on lipid changes become more speculative because they cannot say to which part the lipid composition is driven by the host vs the symbiont.

Authors: We appreciate the overall positive feedback of the reviewer. We agree that being able to separate the two organisms for this purpose would be useful. However, in order to obtain sufficient biomass of the nanohaloarchaeon for lipidomics post-cultivation it was necessary for us to filter a minimum of 200 mL of each co-culture from which we obtained ~40 mg of dry-weight biomass on average. If we were to attempt this for all timepoints we would need to conduct the initial inoculation into >1 L of host culture for each replicate. The filtration procedure is effective in purification of nanohaloarchaea but not particularly efficient and results in the loss of 90% of the nanohaloarchaeal cells. As a result, to provide the nanohaloarchaeal biomass for such an experiment with consistent cell ratios as in the current iteration we would require 30 L of enrichment culture to filter. Using our current protocol and filtration equipment, purification of nanohaloarchaeal cells from this volume would require ~3 weeks (the majority of this time is spent concentrating cells in the filtrate by centrifugation) during which time the initial purified cells would likely begin to display effects from prolonged absence of host cells. It is also not possible to isolate the host from the co-cultures into pure samples as interacting nanohaloarchaeal cells will remain attached. We agree this would have been beneficial, but we hope the reviewer understands that the logistics of handling the system are the reason we opted against performing the experiment in this way.

I wonder about the utility of Fig. 2 to bring the story across. While it is a very pretty figure, I have a hard time assessing the visually only subtle differences in these network plots. Furthermore, as most of the discussion focusses on Fig. 3 this figure does not even seem necessary to me. I would therefore recommend to remove Fig. 2 or move it to the supplement. Similarly, I don't see the necessity to show Figs. 1b and 1c in the main text. Fig. 1 is as it is already jam-packed with information.

Authors: We thank the reviewer for their suggestions. We acknowledge that Figures 2B-C may provide overlapped information as Figure 3A; as they offer a quick visualization to aid in data interpretation, therefore we have included them in the supplementary material. However, we believe Figure 2A is crucial as it conveys information essential for all audiences, especially non-experts, and hence we prefer to retain it as Figure 2. As discussed in the Discussion section, unlike other reported archaeal hosts that primarily have membranes consisting of monolayer membrane tetraether lipids, such as glycerol dialkyl glycerol tetraethers (GDGTs), the cellular membranes of *Hrr. lacusprofundi* are uniquely composed of bilayer AR lipids. This composition in the *Halobacteriales* represents one of the most extreme instances of negatively charged membranes across the tree of life, likely an adaptation to their highly cationic environment. Figure 2A (now Figure 2) offers a quick overview of the lipid compositions of *Ca. Nha. antarcticus* and *Hrr. lacusprofundi*, highlighting the significant difference in lipid types between our symbiont system and others that primarily consist of monolayer membrane tetraether lipids. We have

now made this more clear in the manuscript. With regards to Fig. 1B and C, we feel inclusion of the growth data in the main text is beneficial in providing confirmation that growth states were comparable between the two conditions and that *Ca. Nha. antarcticus* was actively proliferating during the experiment. Due to the capacity for differences in growth state to influence membrane lipid composition we prefer to keep this data in the main text so that it is made clear this is unlikely to have impacted our experiment.

Specific comments.

Line 160: Change to “menaquinones”

Authors: Corrected

Line 354: I would use the word “normalized” instead of “calibrated” since it sounds misleading to write that “peak area responses were calibrated” when in fact they were not.

Authors: Corrected

Line 354: I don't understand the use of the word “necessarily”. The standard DGTS for normalization has structurally very little in common with the investigated lipid compounds, therefore, there is no reason to assume that the use of this standard may have accounted for differences in peak area responses. Thus, I suggest to remove the word “necessarily”.

Authors: We have removed the word ‘necessarily’ as suggested.

Line 438: The ‘S’ should be in upper-case for 16S rRNA

Authors: Corrected

Fig. 1 I find the structure of the figure panels slightly confusing. A, b and c are vertically aligned, but d, e, f and g are horizontally aligned? Is there a purpose for this disorder? I would also advise the authors to not only rely on color when making their plots. It's always helpful to use different symbol types to be considerate of color blind people.

Authors: We appreciate the suggestions provided by the reviewer. The arrangement of the figures was indeed influenced by spatial constraints. For instance, Figures 1b and 1c are relatively small; placing them in the second line would result in excessive empty space, whereas the bottom figures would appear overly congested. We propose to defer the decision regarding the optimal arrangement of these figures to the editorial office's discretion.

Fig. 2: This figure looks fantastic, but I don't see a lot of additional gain by showing this (see comments above).

Authors: Please see our response above.

Fig. 3a: What do the letters a,b,c,d stand for above the individual bars?

Authors: The letters are indicators of statistical significance. This method is commonly used in the scientific literature. If two samples are not statistically significantly different from one another they are assigned the same letter, otherwise they are given different letters. If a sample is not statistically significantly different from two groups then it is assigned both letters (e.g. all pure host cultures in the MK(8:8) plot are assigned 'ab' which means they are not significantly different for either 'a' or 'b'). In essence, any samples with one or more shared letter are not significantly different, any samples with no shared letters are statistically significantly different.

Fig. 3b: I don't understand the x-axis in this plot. What is meant by genre? Please explain this in the figure caption. What do grey dots mean?

Authors: We have changed the term "Genre" from the axis to "number of lipid species". Genre was the default term for describing the cluster of samples that share a set of lipids but we agree is not particularly intuitive. The dots on the x-axis show whether the lipids included in the bar plot are present in the sample or not. A black dot means the sample contains all these lipids, a grey dot indicates it includes none. As an example, the 8th bar shows lipids only found in *Ca. Nha. antarcticus* samples and so the Nha circle is black while all others are grey. The lipids used for producing these plots included lipid species identified in control samples which is why the second bar shows 30 lipid species not found in any of the samples included in the plot.